# Genome-Wide mRNA and Long Non-Coding RNA Analysis of Porcine Trophoblast Cells Infected with Porcine Reproductive and Respiratory Syndrome Virus Associated with Reproductive Failure

**DOI:** 10.3390/ijms24020919

**Published:** 2023-01-04

**Authors:** Xinming Zhang, Xianhui Liu, Jiawei Peng, Sunyangzi Song, Ge Xu, Ningjia Yang, Shoutang Wu, Lin Wang, Shuangyun Wang, Leyi Zhang, Yanling Liu, Pengshuai Liang, Linjun Hong, Zheng Xu, Changxu Song

**Affiliations:** 1College of Animal Science & National Engineering Center for Swine Breeding Industry, South China Agriculture University, Guangzhou 510642, China; 2Lingnan Modern Agricultural Science and Technology Guangdong Laboratory, Guangzhou 510000, China; 3College of Biological Science, University of California-Davis, Davis, CA 95616, USA

**Keywords:** PRRSV, porcine trophoblast cells (PTR2), genome-wide mRNA and long non-coding RNA analysis, GPER1, PI3K-AKT-mTOR

## Abstract

Porcine reproductive and respiratory syndrome (PRRS) is a vertically transmitted reproductive disorder that is typically characterized by miscarriage, premature birth, and stillbirth in pregnant sows after infection. Such characteristics indicate that PRRSV can infect and penetrate the porcine placental barrier to infect fetus piglets. The porcine trophoblast is an important component of the placental barrier, and secretes various hormones, including estrogen and progesterone, to maintain normal pregnancy and embryonic development during pregnancy. It is conceivable that the pathogenic effects of PRRSV infection on porcine trophoblast cells may lead to reproductive failure; however, the underlying detailed mechanism of the interaction between porcine trophoblast (PTR2) cells and PRRSV is unknown. Therefore, we conducted genome-wide mRNA and long non-coding RNA (lncRNA) analysis profiling in PRRSV-infected PTR2. The results showed that 672 mRNAs and 476 lncRNAs were significantly different from the control group after viral infection. Target genes of the co-expression and co-location of differential mRNAs and lncRNAs were enriched by GO (gene ontology) and KEGG (Kyoto Encyclopedia of Genes and Genomes) analysis, revealing that most of the pathways were involved in cell nutrient metabolism, cell proliferation, and differentiation. Specifically, the estrogen signaling pathway, the PI3K (PhosphoInositide-3 Kinase)-Akt (serine/threonine kinase) signaling pathway, and the insulin secretion related to embryonic development were selected for analysis. Further research found that PRRSV inhibits the expression of G-protein-coupled estrogen receptor 1 (GPER1), thereby reducing estrogen-induced phosphorylation of AKT and the mammalian target of rapamycin (mTOR). The reduction in the phosphorylation of AKT and mTOR blocks the activation of the GPER1- PI3K-AKT-mTOR signaling pathway, consequently restraining insulin secretion, impacting PTR2 cell proliferation, differentiation, and nutrient metabolism. We also found that PRRSV triggered trophoblast cell apoptosis, interrupting the integrity of the placental villus barrier. Furthermore, the interaction network diagram of lncRNA, regulating GPER1 and apoptosis-related genes, was constructed, providing a reference for enriching the functions of these lncRNA in the future. In summary, this article elucidated the differential expression of mRNA and lncRNA in trophoblast cells infected with PRRSV. This infection could inhibit the PI3K-AKT-mTOR pathway and trigger apoptosis, providing insight into the mechanism of the vertical transmission of PRRSV and the manifestation of reproductive failure.

## 1. Introduction

Porcine reproductive and respiratory syndrome virus (PRRSV) is classified as *Betaarterivirus suid 1* (European type, PRRSV-1) and *Betaarterivirus suid 2* (American type, PRRSV-2, the strain used in this study), according to the latest classification criteria of the International Committee on Taxonomy of Viruses (ICTV). Both of these strains belong to the genus *Betaarterivirus*, subfamily *Variartevirinae*, family *Arteriviridae* [1]. The main clinical symptoms are miscarriage in pregnant sows, stillbirth, mummified fetuses, weak piglets, and other reproductive disorders, as well as respiratory symptoms and high mortality in piglets [2,3]. Although countries around the world attach great importance to research on PRRSV diagnosis and prevention, etiology, immunology, and molecular biology, areas in which much progress has been made [4,5], the prevention and control of PRRS remains a challenge to the healthy development of the pig industry.

The porcine placental barrier is the site for the exchange of nutrients and metabolites between the mother and the fetus, and the placenta also serves as a natural barrier to prevent various pathogens from entering the fetus through the mother. It is mainly composed of the trophoblast, villous capillary endothelial cells, and a basement membrane [6]. Cells in the pig placenta, especially trophoblast cells, play an extremely important role in embryonic development and can secrete and transport a large number of growth factors and hormones, such as the vascular endothelial growth factor, the epidermal growth factor, the insulin-like growth factor, and estrogen [7]. At the same time, trophoblast cells are also effector cells of these growth factors and hormones, and the recognition of their related receptors further promotes the recognition of pregnancy signals by the mother and embryo implantation, and increases the blood flow and nutrient transport of the placenta [8]. However, a large number of studies have shown that it is also an important means of vertical transmission of pathogenic microorganisms. For example, PRRSV can infect and cross porcine trophoblast cells [9]. Therefore, it may be presumed that these symptoms are related to the infection of trophoblast cells with PRRSV, considering the clinical manifestations of induced abortion, stillbirth, and mummified fetuses in pregnant sows with PRRSV, and the obvious viremia at birth in piglets with intrauterine PRRSV infection.

Previous studies have indicated that PRRSV infection causes a series of changes in the expression of genes that regulate different biological functions, including apoptosis and autophagy of the infected cells [10,11] and the production of inflammatory cytokines [3]. For example, the co-expression of 126 lncRNA and 753 mRNAs was examined in porcine alveolar macrophage (PAM) cells infected by PRRSV; some of them were enriched in the nuclear factor kappa-light-chain-enhancer of activated B cells (NF-κB) and the toll-like receptor (TLR) pathways, and others were related to interferon-induced genes [12]. Furthermore, a comprehensive analysis of the transcripts of microRNAs and mRNAs was conducted for PRRSV-infected Mark-145 cells; some new miRNAs and target genes were found to be related to the pathogenesis of PRRSV and the innate immune response of the host [13]. However, transcriptome analyses that focus specifically on host cells infected by PRRSV mainly consider the respiratory and immune systems, with only a few studies examining the effects of PRRSV on the reproductive system. Porcine endometrial epithelial cells (PECs) on the sow side and trophoblast cells on the fetal side are the two main layers of the porcine placenta. Studies focused on PECs were reported in 2019; they offered whole-transcriptome analysis of PRRSV-infected PECs, and suggested that apoptosis, necroptosis, and the p53 signaling pathways were significantly enriched [14]. However, elucidating the detailed mechanisms of the infection of porcine trophoblast cells by PRRSV require genome-wide analysis of mRNA and lncRNA, and its manifestation remains unclear.

In this study, we performed a genome-wide analysis of mRNA and long noncoding RNA profiles in PRRSV-infected PTR2 cells, and some of the relevant pathways were functionally verified. The purpose of this research is to provide a basis for a more comprehensive understanding of the pathogenic mechanisms related to the vertical transmission and clinical manifestations of PRRSV.

## 2. Results

### 2.1. Cytopathic Effect (CPE) of PTR2 Infected with PRRSV

Our experiments demonstrated that PRRSV can infect PTR2 cells, as shown in Figure 1A. At 72 h of PRRSV infection, the PTR2 cells gathered into clumps, became swollen, round, and fell off, and fused into multinucleated cells. To further verify the infection efficiency, the PTR2 cells were compared with the Mark145 cells, and the gene quantization results showed that the infection efficiency in the PTR2 cells was much lower than that in the Mark145 cells (Figure 1B). The extent of the cytopathic effect (CPE) was different between the two cell lines. At 72 h of virus infection, Mark145 cells showed typical cytopathic changes and a wide range of cells were rounded and shed, while the PTR2 cells only showed sporadic cytopathic changes without shedding. As shown in Figure 1C, at 48 h of virus infection with different doses, it was found that virions significantly increased in a dose-dependent manner, and the nuclear aggregation phenomenon of the cells was also significantly enhanced.

### 2.2. RNA Sequencing Data Processing and Quality Identification

Two groups of samples, including the PRRSV-infected PTR2 cells and the normal control group with three replicates, were collected; thus, a total of six samples (MOCK1 MOCK2, MOCK3, PRRSV1, PRRSV2, and PRRSV3) were subjected to Next Generation Sequencing (NGS, Illumina PE150/SE50). The original image file was designed using CASAVA base identification of the sequence-read section [15]. A small number of reads with sequence joints or low sequencing quality were filtered out and the sequence error rate and GC content distribution were checked out. Approximately 92 million clean reads including mRNA and lncRNA were obtained for each sample. The data quality is summarized in Table 1.

Comparing the clean reads to the genome or transcriptome is the basis of the subsequent analysis. The comparative analysis of the RNA-seq data was conducted using the Hisat2 software. The effective data that are mapped to the reference genome are counted, and Table 1 shows the statistical mapping rate.

The transcript length, ORF length, and exon number of the mRNAs and lncRNAs were compared. The lncRNAs had shorter transcripts and ORFs, and fewer exons than the mRNAs (Figure 2A–C). These data not only show the difference between the lncRNA and mRNA but also verified that the predicted novel lncRNA was in line with the general characteristics. Box plots were used to show the distribution of gene or transcript expression levels in the different samples in Figure 2D. The transcriptional level of the group with PRRSV infection was slightly higher than that of the normal group. 

### 2.3. Expression Profiles of the mRNAs and lncRNAs in the PTR2 Cells with PRRSV Infection

The PRRSV infection altered the transcription profile of the PTR2 cells. About 16,060 novel mRNA and 7776 novel lncRNAs, and a total of 672 mRNAs (406 upregulated and 266 downregulated) and 476 lncRNAs (284 upregulated and 192 downregulated) were found to be significantly different after the viral infection (Figure 3A,B). They were widely distributed among almost all the chromosomes in the genome. The differential lncRNAs and mRNAs were analyzed using hierarchical cluster analysis. The heat map shows significant self-segregated clusters in the control group and the PRRSV-infected PTR2 cell group. (Figure 3C,D).

Subsequently, four mRNAs and two lncRNAs with differential expressions were randomly selected for quantitative gene detection to confirm the accuracy of the RNA-sequencing. The results showed that the mRNAs *BCL2L1, CASP7*, and *CASP9* were significantly upregulated compared to the control group, while the mRNA *CYCS* were significantly down-regulated (Figure 3E). The lncRNA ENSSSCG00000021203 was significantly increased, while the expression of lncRNA ENSSSCG00000000720 was significantly reduced (Figure 3F). These genes remained consistent with the trend of the sequence; however, there were some differences in the regulatory multiplex between the two methods.

### 2.4. Pathway Analysis for the Abnormal Expression of mRNAs and lncRNAs in the PTR2 Cells Infected with PRRSV

To understand the biological functions of the differential genes, the mRNAs and lncRNAs with differences between the PRRSV-infected PTR2 cells group and the normal group were used for the GO and KEGG enrichment analysis. The GO classification analysis showed that the PRRSV infection mainly affected cell proliferation and differentiation, morphological changes in the nucleus and genetic material, cell basal metabolism and RNA transcription, and especially the regulation mechanisms, such as “regulation of transforming growth factor” “RNA polymerase II transcription regulation” “regulation of epidermis development”, “nucleoplasm”, “positive regulation of metabolic process”, and “regulation of RNA-directed RNA polymerization”, (Figure 4A). The co-expressed genes of the abnormally lncRNAs in the PRRSV infection group were significantly reflected in the “RNA metabolic process” “gene expression”, and “cellular metabolic process” (Figure 4B). In addition, the KEGG-analysis-enriched pathways were mainly distributed among “PI3K-Akt signaling pathway”, “ECM-receptor interaction”, “Estrogen signaling pathway”, and “Focal adhesion”, (Figure 4C), among the dysregulated mRNA co-expression genes in the PRRSV-infected group. Furthermore, the “Notch signaling pathway”, “Hippo signaling pathway”, and “AMPK signaling pathway” were enriched in the dysregulated lncRNAs in the treatment group (Figure 4D). The top 20 KEGG pathways and all the co-expressed genes of the dysfunctional lncRNAs and mRNAs are also listed in Appendix A.

### 2.5. PRRSV Inhibits the Expression of GPER1 in PTR2 Cells and Downregulates the Activation of the PI3K-AKT-mTOR Signaling Pathway Induced by Estrogen

As shown in Figure 4C, PRRSV infection of PTR2 cells can regulate the estrogen signaling pathway. It goes without saying that the estrogen signaling pathway plays a crucial role in all stages of embryonic development. The premise of the activation and function of the estrogen signaling pathway is that estrogen is recognized by estrogen receptors. However, we found that PRRSV infection inhibits the expression of GPER1 (Figure 5A,B), and the network diagram of lncRNAs regulating GPER1 mRNA transcription in PTR2 cells infected with PRRSV is shown in Figure 6. Notably, we also found that estrogen inhibited PRRSV replication in a dose-dependent manner (Figure 5B). Studies have shown that estrogen can promote the expression of GPER1 [16], and then activates the PI3K-AKT-mTOR signaling pathway after recognition.

PI3K-AKT-mTOR is a classic pathway that responds to insulin signaling, regulating insulin secretion and instructing cells to absorb and utilize nutrients. Interestingly, the “PI3K-AKT signaling pathway” and “insulin secretion” were significantly enriched in KEGG enrichment analysis (indicated in blue in Figure 4C). Subsequently, we used estrogen as an activator to verify the relationship between PRRSV and the PI3K-AKT-mTOR signaling pathway. As shown in Figure 5C, PRRSV inhibited estrogen-induced AKT and mTOR phosphorylation levels.

### 2.6. PRRSV Induced the Apoptosis of the PTR2 Cells

Some researchers have found that, when the estrogen signaling pathway is inhibited, the Bcl-2 gene’s transcription efficiency is also significantly reduced, which subsequently leads to the apoptosis of a large number of trophoblast cells [17,18]. Transcriptomic results showed that the expression level of the Bcl-2 gene was significantly decreased in PTR2 cells infected with PRRSV, and KEGG enrichment analysis showed that the “apoptosis signaling pathway” was also significantly enriched (not in the top 20). Therefore, further studies are needed to verify this. The flow cytometry results indicated that PRRSV could induce the apoptosis of the PTR2 cells after infection with PRRSV, and the induction effect was more obvious with the increase of the viral titer (Figure 7A). PRRSV was found to accelerate the transformation process of the cells from normal cells undergoing early apoptosis to cells undergoing late apoptosis (Figure 7B).

### 2.7. The Interaction between the Differentially Expressed mRNAs and lncRNAs Associated with Apoptosis in the PTR2 Cells Infected with PRRSV

Since network analysis can provide a better global view of all the possible lncRNAs-mRNAs expression associations based on different apoptotic backgrounds, network analysis was used here to predict the functional annotations of the lncRNAs and to further highlight the relationship of the dysregulation of lncRNAs and mRNAs associated with apoptosis in the PRRSV-infected PTR2 cells. We observed the co-expression interaction between the five protein-coding genes from the apoptosis upstream pathways (*IRAK3, RELA, PIK3R1, TRAF2, NFκB1*) and six protein-coding genes from the apoptosis downstream pathways (*BCL2L1, CYCS, CASP6, CASP7, CASP9, CASP10*) with their predicted regulatory factors lncRNAs (Figure 8A,B). In addition, to facilitate further research, two genes (*BCL2L1, CYCS*) that were highly differentially expressed in the RNA sequencing results were selected to list the network separately (Figure 9A,B). These results provide some basis for revealing the apoptosis of the trophoblast cells and inducing abortion, stillbirth, and mummified birth of the pregnant sows infected with PRRSV.

## 3. Discussion

During the pregnancy of sows, porcine trophoblast cells grow rapidly from day 12 to day 18 of gestation and release hormones such as 17β-estradiol to recognize pregnancy signals. At 13–14 days, trophoblasts differentiate and attach to the uterine cavity epithelium [19,20]. In the later stage, maternal endometrial epithelial cells and fetal trophoblast cells adhere closely and fold into the placenta, providing nutrition and protection for the fetus and ensuring the normal development of the fetus [21]. Therefore, to some extent, the smooth progress of pregnancy completely depends on whether trophoblast cells function normally or not. However, many pathogens of reproductive disorders preferentially attack trophoblast cells in order to break through the placental barrier and infect the offspring. PRRSV, one of the most devastating viruses affecting the stability of the global pig industry, can directly infect trophoblast cells and cause lesions, leading to embryo infection and affecting fetal development [9]. In this study, it was confirmed that PRRSV could infect PTR2 cells and induce typical cytopathic effects in a dose-dependent manner. This may be direct evidence that PRRSV causes reproductive disorders.

Estrogen plays an important role in regulating follicle development, sow estrus, embryonic development, parturition, and lactation. The premise of estrogen functioning is that it is recognized by estrogen receptors. Estrogen receptor alpha (ERα) and estrogen receptor beta (ERβ) are the two main types of classical estrogen receptors, both of which belong to steroid hormone receptors and are localized in the cytoplasm and nucleus. Estrogen enters cells to specifically bind to ERα and Erβ, and produces corresponding biological effects by regulating the expression of target genes in the nucleus. Estrogen also has a rapid non-classical genomic effect, independent of ERα and ERβ, which is mediated by a novel estrogen receptor GPER1 molecule located on the cell membrane [22]. When estrogen specifically binds to the GPER1 molecule, a physiological response can be produced within seconds through the GPER1/EGFR signaling pathway. The core molecules of the GPER1/EGFR signaling pathway mainly include GRER1, epidermal growth factor receptor (EGFR), mitogen-activated protein kinase (MAPK), PI3K, AKT, and mTOR [23,24]. Among these molecules, mTOR acts as a downstream target of PI3K-AKT and MAPK cascades signaling, which can promote the release of insulin after activation, instructs cells to regulate glucose transport and metabolism, and provides energy for the body [25,26]. In addition to this, these molecules are involved in cell proliferation and differentiation [27], apoptosis [28] and autophagy [29]. In this study, transcriptomic analysis of PRRSV-infected PTR2 cells showed that the “estrogen signaling pathway”, the “PI3K-AKT signaling pathway”, and “insulin secretion” were significantly enriched. In-depth studies have found that PRRSV inhibits the expression of the GPER1 gene, thus impairing the binding of estrogen to GPER1, and consequently downregulating the phosphorylation levels of AKT and mTOR, resulting in the blocking of the estrogen-GPER1-EGFR-PI3K-AKT-mTOR signaling pathway, which in turn decreases insulin secretion and reduces the nutrient metabolism. This mechanism provides theoretical support for the theory that PRRSV infection of pregnant sows leads to abortion, weak fetuses, and mummified fetuses. On the other hand, we also found that estrogen can inhibit the proliferation of PRRSV in PTR2 cells. This means that, for PRRSV-infected sows with abnormal estrus and pregnancy, the appropriate supplementation of estrogen can not only alleviate the effect of PRRSV on the suppressed expression of GPER1, but also inhibit the further proliferation of PRRSV.

Several studies have shown that PRRSV infection of PAM cells and MARC-145 cells can cause apoptosis to varying degrees [30,31]. Other studies have found that, when the estrogen signaling pathway is inhibited, trophoblast cells show a high degree of apoptosis [18]. In this study, apoptosis detection was performed, and the PRRSV was found to induce the apoptosis of the PTR2 cells in a dose-dependent manner. This result implies that trophoblast cells are destroyed by PRRSV infection. The destruction of the trophoblasts not only results in decreased embryo colonization, but also damages the integrity of the placental villus barrier and leads to embryonic cessation, which may also be the cause of PRRSV-induced abortion in pregnant sows. Some studies have shown that the PI3K/Akt pathway is positively regulated by PIK3R1 inducing apoptosis [32]. Moreover, lnc-IRAK3-3 captures miR-891b, enhancing the expression of GADD45β and promoting cell apoptosis [33]. In order to better understand the regulation of lncRNA and mRNA in the PRRSV-induced apoptosis of PTR2 cells, the mutual regulatory interaction between the key mRNAs and lncRNAs enriched in the apoptosis pathway was divided into the upstream and downstream pathways, respectively, to produce network diagrams. *TRAF2, PIK3R1, IRAK3, NFκB1*, and *RELA* were found to be relatively important signaling pathway nodes. In our study, co-expression and co-location prediction analysis identified many lncRNAs that are involved in regulating the expression of these mRNAs, and these new regulatory mechanisms need further validation. BCL2L1, CYCS, and cysteinyl aspartate-specific proteinases were involved in apoptosis [32,33,34]. We have separately listed all the lncRNA network diagrams that regulate the expression of the BCL2L1 and CYCS genes which showed large expression differences, providing the basis for further research.

In this study, in addition to the pathways that are widely enriched in the dysregulation of the mRNAs in the other PRRSV-infected cell lines, some special pathways attracted our attention. For example, the “Focal adhesion” and “ECM-receptor interaction” were also mentioned (Top 20). Studies have shown that the multifunctional focal adhesion complexes promote the contact of the extracellular matrix and the connection between the extracellular matrix and the actin skeleton. They play a key role in the mechanism because they control the morphology of the cells and the cytoplasmic signal transduction for cell survival, proliferation, differentiation, and apoptosis in structure and function [35,36,37]. This provides another insight into the pathogenesis of PRRSV. Of course, these pathways still need further validation. In the enriched lncRNA pathway, the “Notch signaling pathway” was enriched. Studies have shown Notch to encode a class of highly conserved cell surface receptors that play important functions in the cell development of various organisms, from sea urchins [38] to humans [39]. Several processes of cell morphogenesis are regulated by the Notch signaling pathway, including cell boundary formation, multipotent progenitor cell differentiation, cell proliferation, and cell apoptosis [40,41]. The expression of the inflammatory cytokines in the porcine alveolar macrophages after infection with highly pathogenic porcine reproductive and respiratory syndrome virus has also been found to be regulated by the Notch signaling pathway [42]. Therefore, we have reason to believe that the abnormal expression of the lncRNAs of this pathway in the porcine trophoblast cells infected with PRRSV is inextricably related to embryonic development. On the other hand, the “Hippo signaling pathway” is known to respond to a variety of extracellular and intracellular signals, from intercellular contact and mechanical signals to ligands and metabolic pathways for the GPER1. Studies have proven that the Hippo signaling pathway can coordinate cell proliferation, cell death, and cell differentiation, and regulate tissue growth as a highly conserved growth-control signaling pathway [43,44,45]. In general, PRRSV infection of the PTR2 cells can activate a variety of pathways to change cell proliferation, differentiation, apoptosis, and other processes.

## 4. Material and Methods

### 4.1. Cell Cultures and Viral Infection

The PTR2 cells were donated by the National Seed Industry Engineering Research Center. A DMEM medium with 10% fetal bovine serum, 1% insulin, and 1% penicillin-streptomycin was added to the cell culture, at 37 °C with 5% CO_2_. The PRRSV (GD-ZJ, GenBank: MF772778.1, American type) was isolated and preserved in our laboratory. The PRRSV stock titers were 1 × 10^7^ TCID_50_/mL.The cells were infected with the virus at the set dose, incubated at 37 °C for 1 h, washed with PBS, replaced with DMEM of 2% FBS; this continued until the samples were collected. The cell culture and infection experiments were carried out in our physical containment level 2 (P2) laboratory.

### 4.2. Immunofluorescence Antibody Assay (IFA)

The PTR2 cells were seeded on 12-well plates with 10% FBS in fresh DMEM and cultured overnight at 37 °C with 5% CO_2_. When the cells grew to 80–90%, they were infected with PRRSV with MOI = 1. The cells were washed with preheated PBST (PBS with 0.5% TWEEN) 2 times, fixed in 4% para-formaldehyde for 10 min, rinsed in PBST 5 times, 5 min each time. They were rinsed twice for 20 min by 0.5% Triton X-100 PBS and then incubated in 1% BSA (Bovine Serum Albumin) at 37 °C for 60 min. The antibody of PRRSV N protein was incubated in a shaker at room temperature (1:200 dilution), incubated overnight at 4 °C, and rinsed with PBST 5 times for 5 min each time. The fluorescein-labeled secondary antibodies (CoraLite 488-conjugated Goat Anti-Mouse lgG (H + L), Proteintech, China) were added with 1:600 dilution, incubated at 37 °C in dark for 50 min, and rinsed 5 times with PBS by 5 min each time. The plates were sealed with an anti-fluorescence quenching agent containing DAPI, and then observed under a fluorescence microscope.

### 4.3. RNA Extraction and Qualification

The Trizol reagent (Sigma) was used to extract the total RNA of the PTR2 cells, then the potential genomic DNA contamination was removed with DNA enzymes, and a 1% agarose gel electrophoresis was used to detect the degradation and contamination of the RNA sample. Subsequently, a NanoPhotometer^®^ spectrophotometer (IMPLEN, Westlake Village, CA, USA) was used to detect the purity of the RNA, and the RNA integrity was accurately measured using an Agilent 2100 BioAnalyzer.

### 4.4. Preparation of the RNA Library and Transcript Sequencing

First, the ribosomal RNA was removed from the total RNA; then, the RNase R was used to break the RNA into short fragments, 250–300 bp in size. Using these fragments as templates and random oligonucleotides as primers, the first cDNA strand was synthesized. Then, the RNase H was used to degrade the RNA strand. The second strand of cDNA was synthesized from dNTPs (dUTP, dATP, dGTP, and dCTP) under the DNA polymerase I system. The purified double-stranded cDNA ends were repaired, the A-tails were added, the sequencing joints were connected, and about 350 to 400 bp cDNA was screened using the AMPure XP Beads. The second strand of cDNA containing U was degraded by the USER enzyme and the PCR amplification was performed to obtain the library.

### 4.5. Sequencing Data Analysis

After inspecting the library, Illumina PE150 sequencing was conducted according to the effective concentration of the library and the pooling of the data output requirements. The PE150 (pair-end 150 bp) refers to high-throughput double-end sequencing, and each end was measured at 150 bp. In the constructed small fragment library, the Insert cDNA (insert fragment) is the unit of direct sequencing. The dual-ended sequencing method was used to sequence both ends of each inserted fragment. Since the length distribution of the inserted fragment is known, the dual-ended sequencing could obtain not only the sequence at both ends of the fragment but also the length between the two sequences, which facilitated the subsequent assembly and comparison.

### 4.6. Differential Expression Analysis and Target Gene Prediction

The clean reads were compared to the genomes and transcriptome; Adopt Tophat 2 [46], Hisat 2 [47], and STAR software [48] were used to conduct a comparative analysis of the RNA sequencing data. We removed the resulting original reads from those containing low-quality reads and adapter sequences to obtain clean reads, which were mapped to the pig genome using TopHat software (Sus scrofa 10.2). The RPKM method was then used to calculate the gene expression level. For differential expression analysis, Benjamini and Hochberg’s methods were used to control the error discovery rate. Genes with fold-change (FC) ≥ 0, *p*-value < 0.05 were identified as differentially expressed (DEG).

### 4.7. Quantitative Real-Time PCR (qPCR) Detection

The HiPure Total RNA Mini kit (Magen, China) was used to extract the total RNA of the PTR2 cells. The cDNA was reverse-transcribed from the RNA by Evo M-MLV RT Premix for qPCR (Accurate Biology, China). The Applied Biosystems StepOnePlusTM System was used to perform the quantitative real-time PCR (qPCR). The specific primers were designed to detect PRRSV and other genes (Table 2).

### 4.8. GO and KEGG Enrichment Analysis

GOseq software was used for the GO enrichment analysis. The pathway enrichment analysis was performed using KOBAS 2.0 software. The GO enrichment was characterized by a significant enrichment (Q value < 0.05); the KEGG enrichment was determined by the Rich factor, which was measured by the Q-value and the number of genes enriched in this pathway. The Q value range was [0, 1], with more significance indicated by an enrichment result closer to zero.

### 4.9. Cell Apoptosis Analysis

The apoptosis of the PTR2 cells infected with PRRSV was detected using an Annexin V-FITC/PI Fluorescent Apoptosis assay kit (Elabscience^®^). The PTR2 cells were treated with 5 μM carbonyl cyanide m-chlorophenyl hydrazine (CCCP) for 30 min as a positive control and the untreated cells were used as a negative control. The experimental cells infected with different doses of PRRSV were cultured for 48 h, washed twice with PBS, and stained with 1 µL PI (1 mg/mL) and 5 µL Annexin V-FITC at room temperature for 20 min; they were then analyzed by flow cytometry within 30 min.

### 4.10. Western Blot

Protease inhibitors and a lysis buffer were added to the treated cells, and the supernatant was removed after centrifugation at a low temperature to prepare the protein samples. The proteins were separated in 12% SDS-PAGE protein gel at 60 V for 30 min, separated at 100 V for 90 min for electrophoresis, and then transferred to the PVDF membrane in an ice bath (Thermal Fisher Science). After blocking with 5% skim milk for 1 h at room temperature, the membrane was incubated with the antibody overnight at 4 °C, washed 5 times for 5 min each with PBS containing 0.05% Tween-20 on a shaker. Then, the second antibody was added to the PVDF membrane and incubated at 37 °C for 1 h, followed by further washing as described above. Subsequently, the images were developed using a Western blot developer and analyzed using Image J software. The antibodies used in this article were as follows: the PRRSV nucleocapsid protein antibody (GeneTex, Irvine, CA, USA), AKT antibody (GeneTex, USA), the Phospho-AKT antibody (Ser473), the GPER1 antibody (GeneTex, USA), the mTOR antibody (Affinity, USA), the Phospho-mTOR (Ser2481) antibody (Affinity, USA), and the Actin antibody (Affinity, USA), as well as goat anti-mouse lgG and goat anti-rabbit lgG (Abcam, Cambridge, UK).

## 5. Conclusions

In this study, we verified that PRRSV can infect PTR2 and produce cytopathic effects; moreover, transcriptomic analysis of PRRSV-infected PTR2 cells found that 672 mRNAs and 476 lncRNAs were significantly different compared to the control group. These differential gene functions are mainly enriched in cell growth and development, proliferation and differentiation, physiological metabolism, apoptosis, and autophagy. Further research found that PRRSV inhibits the expression of GPER1, thereby reducing the estrogen-induced phosphorylation of AKT and mTOR, blocking the activation of the GPER1-EGFR-PI3K-AKT-mTOR signaling pathway, thereby restraining insulin secretion, impacting PTR2 cell proliferation, differentiation and nutrient metabolism. Finally, PRRSV also induces apoptosis in PTR cells. These results provided a basis for revealing the pathogenesis of the reproductive dysfunction caused by PRRSV.

## Figures and Tables

**Figure 1 ijms-24-00919-f001:**
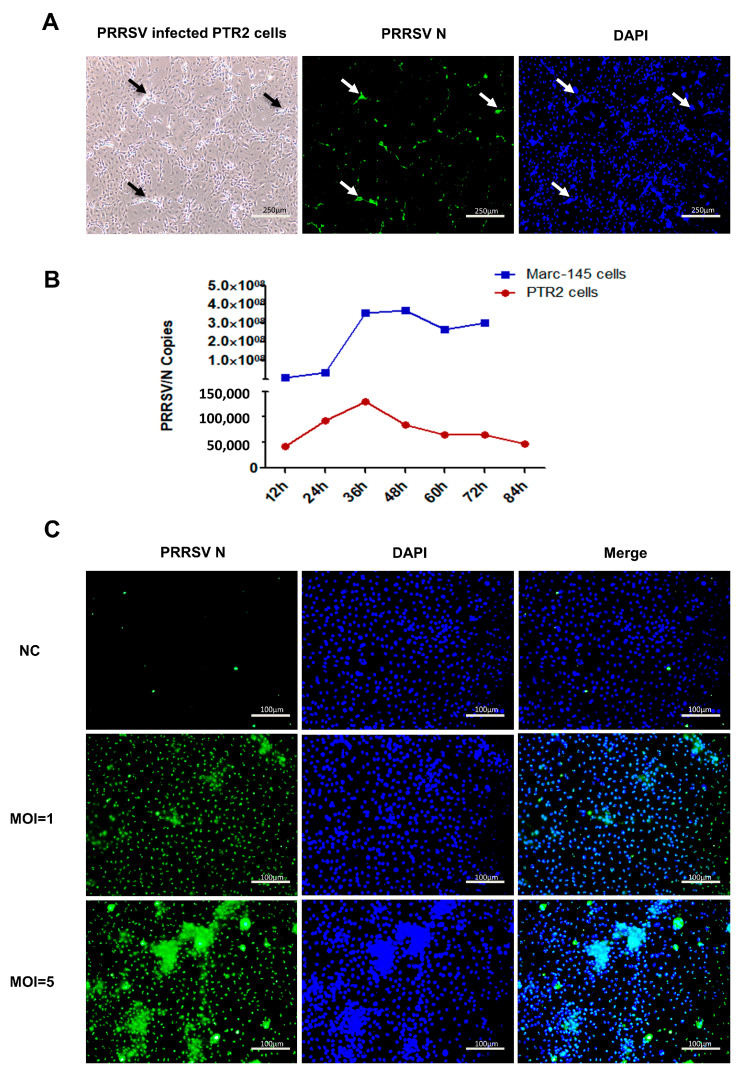
PRRSV infects the PTR2 cells. (**A**) The cell morphology and nuclear morphology of the PTR2 cells infected with PRRSV for 72 h. (**B**) PRRSV with MOI = 1 was inoculated in the MARC-145 cells and PTR2 cells, respectively, and the copy number of the N protein of PRRSV was detected by real-time fluorescence quantification at different time points. The susceptibility and viral load differences of the two cell lines were compared using a line graph. (**C**) The immunofluorescence assay was used to observe the expression of viral N protein in the PTR2 cells infected with PRRSV with MOI = 1 and MOI = 5 at 48 h.

**Figure 2 ijms-24-00919-f002:**
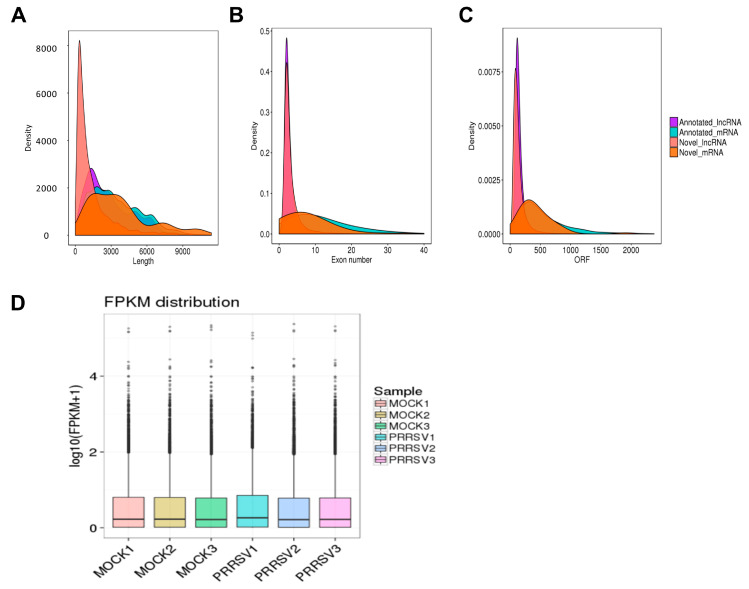
The RNA sequencing and lncRNA identification. (**A**) lncRNA and mRNA length comparison density distribution map. (**B**) Exon number density distributions of lncRNA and mRNA. (**C**) ORF length density distributions of lncRNA and mRNA. (**D**) Boxplots demonstrating the expression levels of lncRNAs and mRNAs in each sample.

**Figure 3 ijms-24-00919-f003:**
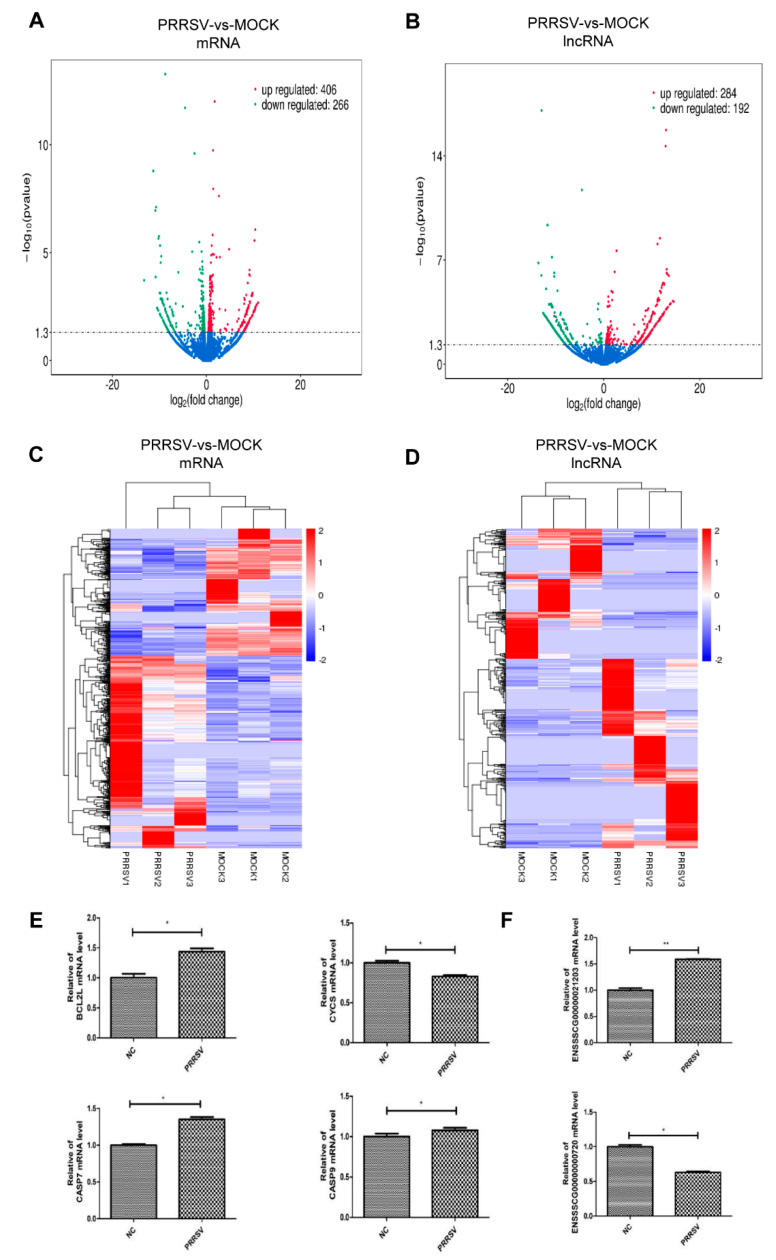
Expression profiles of mRNAs and lncRNAs in the PTR2 cells with PRRSV infection. Volcano plots of the differentially expressed mRNAs (**A**) and lncRNAs (**B**) transcripts. Hierarchical clustering of the expression profiles of differentially expressed mRNAs (**C**) and lncRNAs (**D**). The mRNAs expression of *BCL2L1, CASP7*, and *CASP9* was significantly up-regulated and the mRNA expression of *CYCS* was significantly down-regulated in the PRRSV-infected PTR2 cells compared to the control cells (**E**). ENSSSCG00000021203 lncRNA expression was significantly up-regulated, and ENSSSCG00000000720 lncRNA expression was significantly down-regulated in the PRRSV-infected PTR2 cells compared to the control cells (**F**). * *p* < 0.05, ** *p* < 0.01.

**Figure 4 ijms-24-00919-f004:**
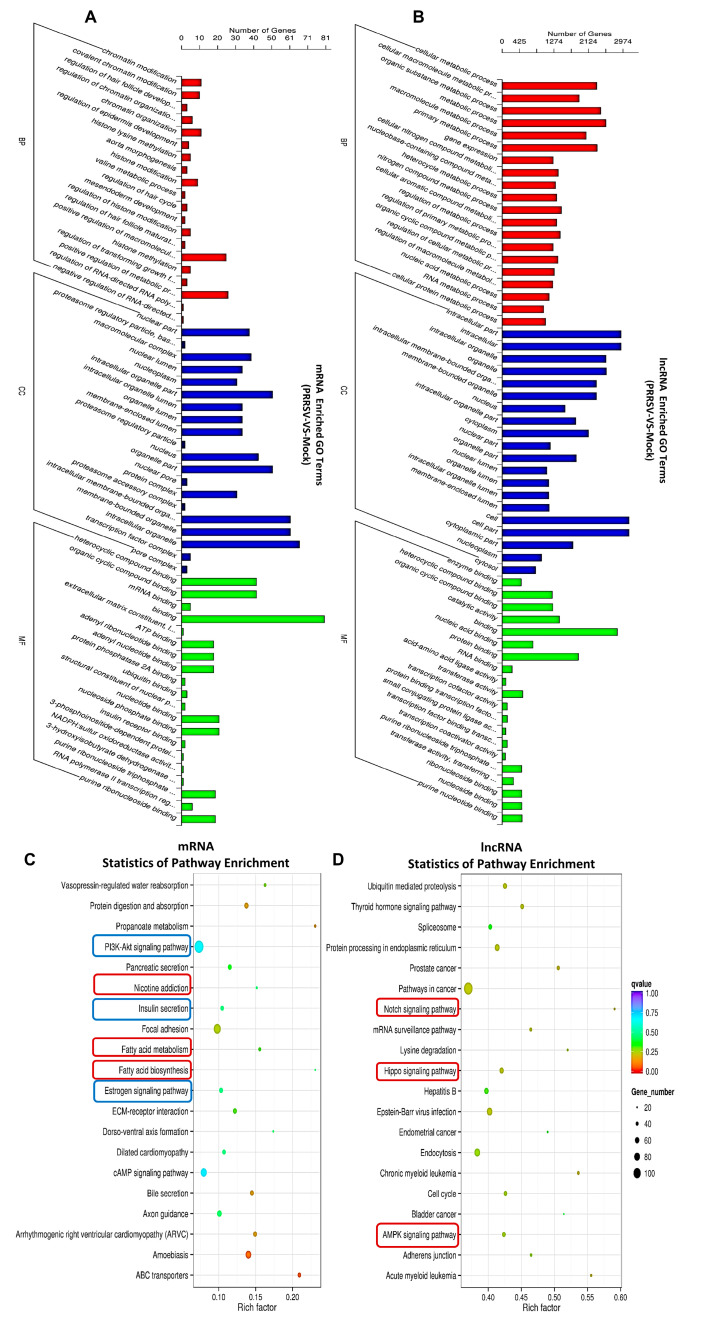
Pathway analysis for the dysregulated mRNAs and lncRNAs in the PTR2 cells infected with PRRSV. Significant enrichment of the GO terms of mRNAs (**A**) and lncRNAs (**B**). Analysis of the top 20 over-represented KEGG pathways of mRNAs (**C**) and lncRNAs (**D**). The enrichment factor was calculated as the number of enriched genes divided by that of all the background genes in each pathway. (**B**,**D**) show the enriched pathways of lncRNAs with effects in the trans-regulated genes.

**Figure 5 ijms-24-00919-f005:**
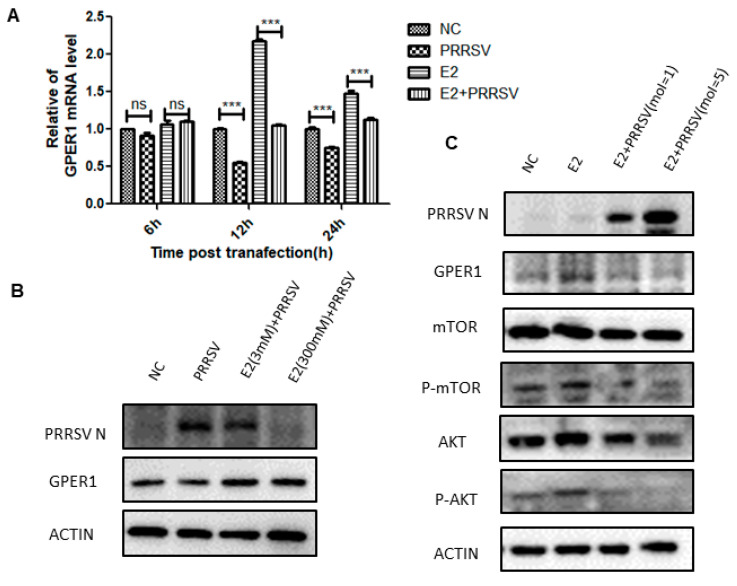
Effects of PRRSV-infected PTR2 cells (MOI = 1) on the transcription of GPER1 genes under estrogen (E2)-stimulated and unstimulated conditions (**A**). Effects of PRRSV infection on PTR2 cells (MOI = 1) on GPER1 protein expression, and the effects of different concentrations of estrogen on PRRSV replication (**B**). Effects of PRRSV in different doses infecting PTR2 cells on the estrogen-induced activation of the PI3K-AKT-mTOR signaling pathway (**C**). *** *p* < 0.001.

**Figure 6 ijms-24-00919-f006:**
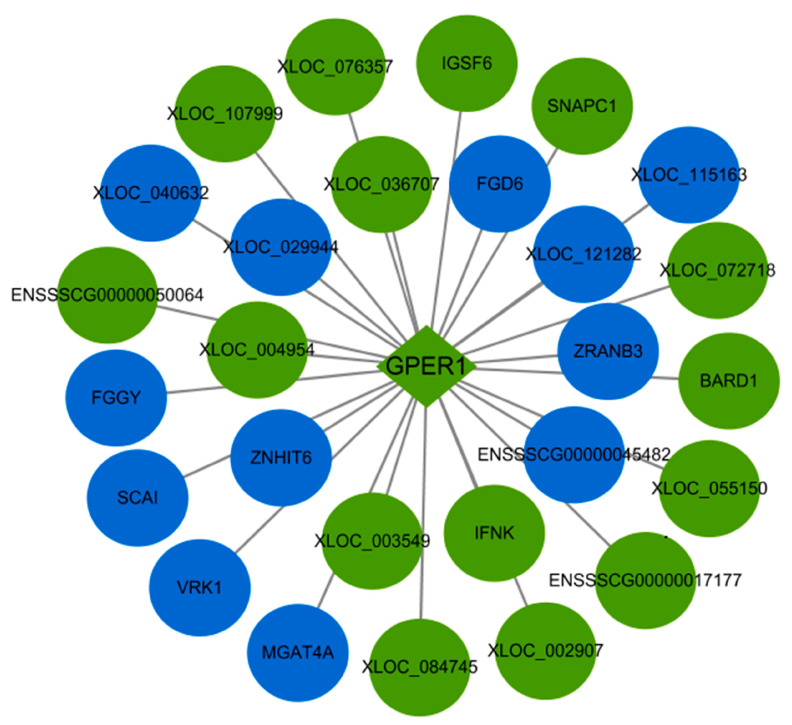
The interaction between the differentially expressed mRNAs and lncRNAs associated with GPER1 in the PTR2 cells infected with PRRSV. Blue means up-regulation, green means down-regulation.

**Figure 7 ijms-24-00919-f007:**
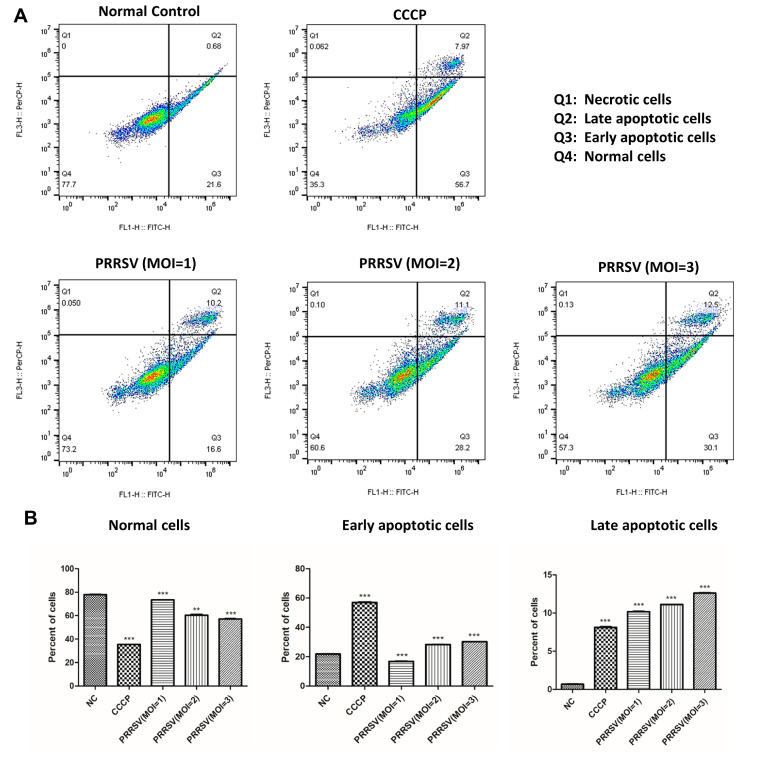
PRRSV induced the apoptosis of the PTR2 cells. Apoptosis ratio analysis of the PTR2 cells with different treatments. The 5 μM carbonyl cyanide m-chlorophenyl hydrazine (CCCP) for 30 min was used as the positive control and untreated cells were used as the negative control. The experimental group was infected with different doses of PRRSV (**A**). Bars show the ratio of normal cells, early apoptotic cells, and late apoptotic cells in each group (**B**). ** *p* < 0.01 and *** *p* < 0.001.

**Figure 8 ijms-24-00919-f008:**
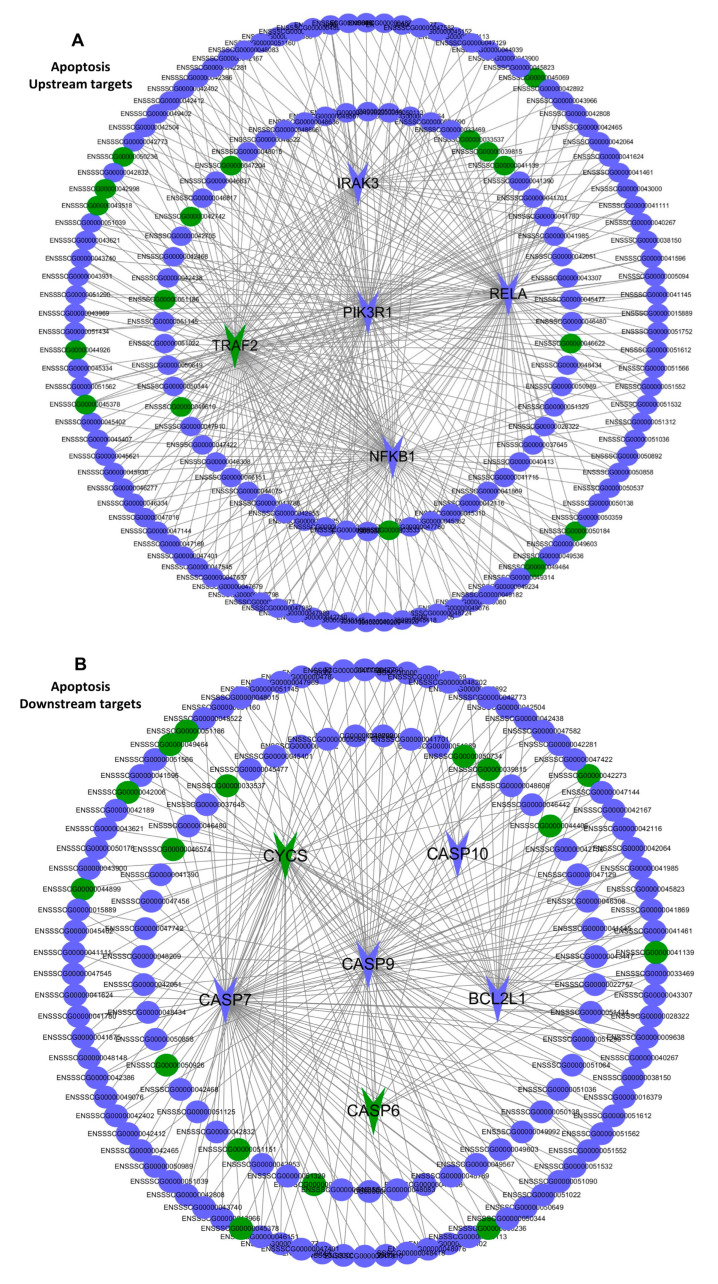
The interaction between the differentially expressed mRNAs and lncRNAs associated with apoptosis in the PTR2 cells infected with PRRSV. Network analysis provides a global view of all the possible lncRNA- mRNA expression associations based on the different upstream targets of apoptosis backgrounds (**A**) and downstream targets of apoptosis backgrounds (**B**) in the PRRSV-infected PTR2 cells. Upstream targets of mRNAs include TRAF2, NFKB1, RELA, PIK3R1, and IRAK3. Downstream targets of mRNAs include CYCS, BCL2L1, CASP6, CASP7, CASP9, and CASP10. Other targets are lncRNAs, which are predicted to regulate mRNA expression. The arrows represent mRNAs and the ellipses represent lncRNAs. Blue means up-regulation, green means down-regulation.

**Figure 9 ijms-24-00919-f009:**
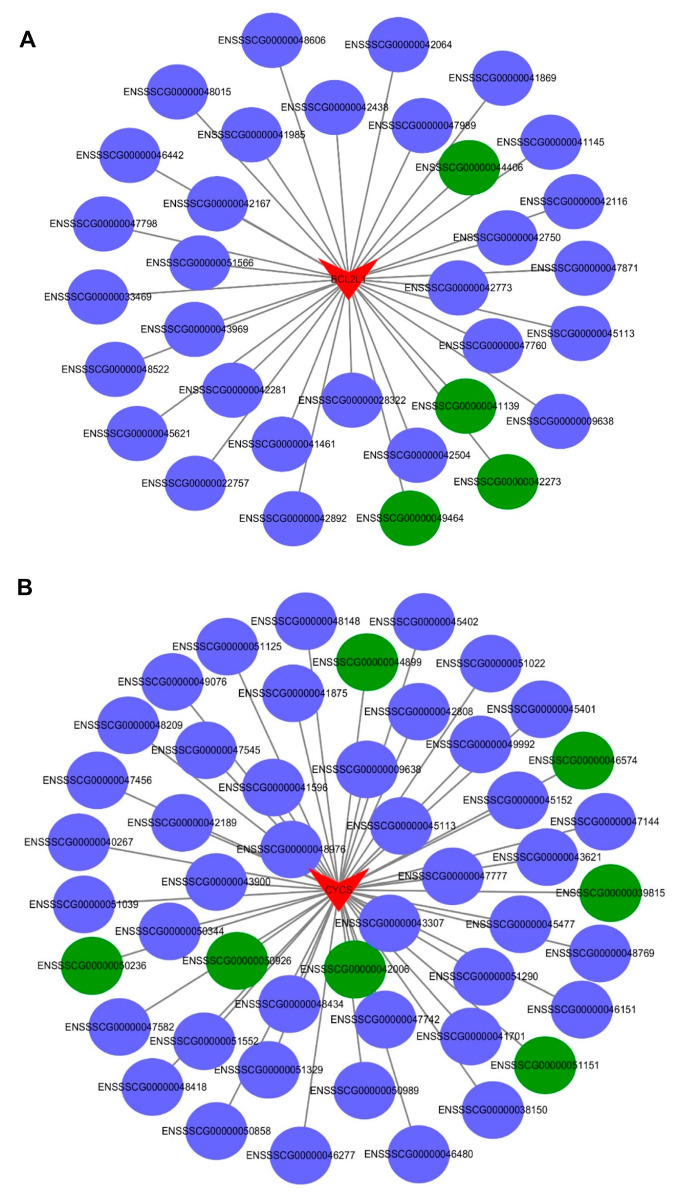
The interaction between the differentially expressed mRNAs and lncRNAs associated with apoptosis in the PTR2 cells infected with PRRSV. BCL2L1 (**A**) and CYCS (**B**) network analysis provide a global view of all possible lncRNA-mRNA expressions. The arrows represent mRNAs and the ellipses represent lncRNAs. Blue means up-regulation, green means down-regulation.

**Table 1 ijms-24-00919-t001:** Overview of the RNA sequencing data.

Sample	Raw Reads	Clean Reads	Q20 (%)	Q30 (%)	GC Content (%)	Total Mapped (%)
Mock 1	93772488	91548686	98.17	94.83	54.04	86282174 (94.25%)
Mock 2	93224184	89721222	98.16	94.81	54.85	85032843 (94.77%)
Mock3	89685878	87053146	98.16	94.80	54.50	81428341 (93.54%)
PRRSV1	91868782	88302932	97.78	94.06	53.83	76908982 (87.1%)
PRRSV2	96897448	94476562	98.32	95.14	52.24	81424721 (86.19%)
PRRSV3	107694804	105835468	98.31	95.14	52.59	91366566 (86.33%)
Average	95523931	92823003	98.15	94.80	53.68	83740605 (90.36%)

**Table 2 ijms-24-00919-t002:** Primers used in this study.

Genes	Forward Sequence (5;-3′)	Reverse Sequence (5;-3′)
PRRSV	CCAGTCAATCAGCTGTGCCA	GACAGGGTACAADTTCCAGCG
BCL2L1	TTGAACGAACTCTTCCGGGA	GTTCTCCTGGATCCAAGGCT
CYCS	GTTCAGAAGTGTGCCCAGTG	CATCAGTGTCTCCTCTCCCC
CASP7	CCGGATGACTCAGACATGGA	TTTCCTGTTCCTCCCCTGAC
CASP9	CCGATTTGGCTTACGTCCTG	CAAAGCCTGGACCATTTGCT
ENSSSCG00000021203	TCCACCAGAGCATGAACCAT	CTCCCACTGACTTGCAACAC
ENSSSCG00000000720	CCCAAGCCAACTAAGGAGGA	AGACACAGCTTCATCACCGA
GPER1	CTTCCTGTCCTGCGTCTACA	GTCGTAGTACTGCTCGTCCA
β-actin	CCGAGATCTCACCGACTACC	CTCGTAGCTCTTCTCCAGGG

## Data Availability

The datasets produced and/or analyzed during the current study are available from the corresponding author on reasonable request.

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
