# Peer review of "Genome-Wide mRNA and Long Non-Coding RNA Analysis of Porcine Trophoblast Cells Infected with Porcine Reproductive and Respiratory Syndrome Virus Associated with Reproductive Failure"

_ijms, 2023, doi:10.3390/ijms24020919_

Round 1

Reviewer 1 Report

Dear Authors,

You really worked hard on this manuscript. It contains lots of data about PRRSV and PTR2 infection and explanations to possible consequences related to Pig Industry. But I noticed multiple errors in the typesetting, methodology order, figures and their captions, etc. You well find all my corrections in the attached file; I used the commenting feature of Adobe Acrobat to annotate all required corrections (41) to your MS and I made also (2) bookmarks for figures 2 and 4 to be considered.

Revising ICTV classification of family Arteriviridae, I noticed that there are 2 types of PRRSV (1 and 2) could you please clarify the type of your virus and update classification accordingly.

I also suggest changing the manuscript title into "Genome-wide mRNA and long non-coding RNA analysis of porcine trophoblast cell (PTR2) infected with Porcine Reproductive and Respiratory Syndrome Virus Associated with reproductive failure" 

Author Response

Dear reviewer,

    We are very grateful for your comments and revisions regarding our manuscript. All your suggestions are very important to us, both for composing the manuscript and our further research. We have studied comments carefully and have made corrections which we hope meet with approval.

    First of all, we have carefully studied and corrected the errors you pointed out in the  typesetting, methodology order, figures and their captions, etc. Your revision is very meticulous and accurate. We have completely improved our article according to your suggestions and uploaded it to the attachment.

    Secondly, The causative agent -PRRSV- is a small, enveloped single-stranded RNA virus that was originally isolated in Europe and North America in the beginning of the 1990s, and in 2018 these two types were classified as two separate species: Betaarterivirus suid 1 (former PRRSV-1, European type) and Betaarterivirus suid 2 (former PRRSV-2, American type) which belong to the genus Betaarterivirus, subfamily Variartevirinae family Arteriviridae.The PRRSV strain (Genebank: MF772778.1) used in our article is an American strain independently isolated by our laboratory, and we have explained it in the article according to your suggestion.

    Finally, as for your suggestion on the title of the article, we have found through discussion that the title you suggested can indeed better describe the full content compared with our title, and we have revised it in the article.

    Thank you again for your selfless help to the improvement of our article. If you have any other suggestions on our article, please do not hesitate to contact us. We look forward to your reply.

                                                                                                    Kind regards,

                                                                                                   Changxu Song

Reviewer 2 Report

1. Some genetic concepts must be reviewed (i.e. mRNA genome, the correct terminology is transcriptome).

2. Bioinformatics methodology should be better described, aiming for the study's reproducibility. Please rearrange the order of the methods described, presenting the RNA-Seq before qPCR.

3. What were the criteria for gene significance (logFC and P-Value)? Was it performed a P-Value adjustment for multiple comparisons?

4. What were the criteria for Gene Ontologies selection?

Author Response

Dear reviewer,

    We have studied your comments carefully. Thank you very much for your suggestions. All your suggestions are very important for the revision of our article. According to your detailed suggestions, we have made a careful revision on the original manuscript. The following is our response to your comment.

    Frist of all, in response to your question about our wrong description and understanding of the concept of genetics, we have consulted some materials for learning. We do have many problems in these fields that we are not very good at. We have reviewed our article and made our best efforts to revise it.The revised parts are underlined below for your further review.

    Secondly, In response to your finding that our bioinformatics methodology is not fully described, our paper further supplements the bioinformatics methodology in materials and methods according to your suggestions, and also marks it in the form of underlining. As for the order of RNA-seq and qPCR you mentioned, we have also made adjustments. Your suggestions on these two points are very valuable for the improvement of our paper.

    Thirdly, Your question "What were the criteria for gene significance (logFC and P-Value)? Was it performed a P-Value adjustment for multiple comparisons?" Our answers are as follows: The difference screening threshold we used for lncRNA and mRNA was pval less than 0.05, and the absolute value of log2FC was greater than 0. In other words, pval less than 0.05 was a significant difference result, and the difference multiple was not adjusted. In the analysis, we also made multiple hypothesis testing correction for pval, and padj values were also provided in the difference results. We made the adjustment by referring to the following literature:

2013 PLOS one - RNA-Seq Profiling Reveals Novel Hepatic Gene Expression Pattern in Aflatoxin B1 Treated Rats (FC>2 & p<0.005)
http://www.plosone.org/article/info%3Adoi%2F10.1371%2Fjournal.pone.0061768
2013 PLOS one - Response of Burkholderia cenocepacia H111 to Micro-Oxia(p<0.2)
http://www.plosone.org/article/info%3Adoi%2F10.1371%2Fjournal.pone.0072939
2013 PLOS one RNA-Seq Differentiates Tumour and Host mRNA Expression Changes Induced by Treatment of Human Tumour Xenografts with the VEGFR Tyrosine Kinase Inhibitor Cediranib(FC>2 & p<0.1)
http://www.plosone.org/article/info%3Adoi%2F10.1371%2Fjournal.pone.0066003
2013 BMC Genomics - Analysis of porcine adipose tissue transcriptome reveals differences in de novo fatty acid synthesis in pigs with divergent muscle fatty acid composition(FC>1.2, p<0.01/q<0.1)
2013 BMC Genomics - Transcriptional profiling of bud dormancy induction and release in oak by next-generation sequencing(q<0.2)
http://www.biomedcentral.com/1471-2164/14/236
2013 BMC Genomics - Transcriptome analysis of the parasite Encephalitozoon cuniculi: an in-depth examination of pre-mRNA splicing in a reduced eukaryote(p<0.01)

 http://www.biomedcentral.com/1471-2164/14/207
2013 BMC Genomics - The transcript catalogue of the short-lived fishNothobranchius furzeri provides insights into age-dependent changes of mRNA levels(p<0.01)

 http://www.biomedcentral.com/1471-2164/14/185
2013 BMC Genomics - De-novo assembly and characterization of the transcriptome of Metschnikowia fructicola reveals differences in gene expression following interaction withPenicillium digitatum and grapefruit peel(p<0.05)
http://www.biomedcentral.com/1471-2164/14/168
2013 BMC Genomics - Comparison and contrast of genes and biological pathways responding to Marek’s disease virus infection using allele-specific expression and differential expression in broiler and layer chickens(p<0.05)
http://www.biomedcentral.com/1471-2164/14/64

    Finally, Your question "What were the criteria for Gene Ontologies selection?” Our answers are as follows: Gene Ontology (http://www.geneontology.org/) is the international standard classification system Gene function. GO can be divided into three parts: molecular function, biological process and cell composition. GO enrichment analysis method is GOseq, which is based on Wallenius non-central hyper-geometric distribution. Compared with ordinary Hyper-geometric distribution, The characteristic of this distribution is that the probability of selecting an individual from a certain category is different from that of selecting an individual from outside a certain category. The difference of this probability is obtained by estimating the bias of gene length, so that the probability of GO term being enriched by the source gene can be more accurately calculated. In GO enrichment, padj less than 0.05 (default) was considered as significant enrichment. We referred to the following literature:

2013 PLOS one -Intrinsic Features in  MicroRNA Transcriptomes Link Porcine Visceral Rather than Subcutaneous  Adipose Tissues to Metabolic Risk. http://journals.plos.org/plosone/article?id=10.1371/journal.pone.0080041

2015 -Scientific Reports- Identification of  volatile and softening-related genes using digital gene expression profiles  in melting peach. http://link.springer.com/article/10.1007/s11295-015-0891-9

2015 RSC Advances -Integrated analysis  of miRNA and mRNA expression profiles in development of porcine testes.

https://www.sci-hub.ren/10.1039/c5ra07488f

2015 PLos ONE-Transcriptomic  Analysis of Ovaries from Pigs with High And Low Litter Size.

http://journals.plos.org/plosone/article?id=10.1371/journal.pone.0139514

   Thank you again for your valuable comments on our article, your guidance is very helpful to us, let us learn a lot of things. Hope our answer can get your recognition, if you still have questions about our article, please do not hesitate to contact us. Looking forward to hearing from you.

    Kind regards,

                                                                                                       Changxu Song

Author Response

Dear reviewer,

   Thank you very much for your suggestions. Your comments are all valuable and very helpful for revising and improving our paper. as well as the important guiding significance to our researches. The following is our response to your comment. 

    First of all, regarding your question about whether there is a causal relationship between the genes we screened and reproductive failure, we would like to make the following additions and explanations: Although we did not verify this in live animals, in trophoblast cells, an important component of pig placenta, we found that the estrogen signaling pathway was significantly different after PRRSV infection. There is no doubt that estrogen signaling pathway plays a very important role in regulating placental development, pregnancy maintenance, fetal growth and other reproductive activities. Subsequently, we have verified that the regulation of estrogen signaling pathway by PRRSV is realized by inhibiting G protein-coupled estrogen receptor 1 (GPER1), and the combination of estrogen and GPER1 is an important link in the downstream signaling molecule expression mediated by estrogen. A large number of studies have shown that the inactivation of GPER1 will terminate fetal development and promote fetal death, which is consistent with the reproductive disorder symptoms caused by PRRSV. In addition, PRRSV infection of trophoblast cells will further cause cytopathy and apoptosis, etc. Therefore, there is every reason to believe that the genes we screened have a causal relationship with PRRSV-induced reproductive disorders. Of course, more sufficient evidence is needed to be demonstrated, and we are still studying further.

   Secondly, Your proposed modification of the title of this paper is consistent with that of another reviewer. We are deeply aware of the defects in our title and have made the following modifications after discussion: Genome-wide mRNA and long non-coding RNA analysis of porcine trophoblast cell infected with Porcine Reproductive and Respiratory Syndrome Virus associated with reproductive failure. If you think it is not perfect enough, you can tell us, we are making further revision.

   Third, about the problem of incorrect English expression you mentioned. Your simple examples have pointed out many mistakes to us, and let us deeply realize our lack of English language expression ability. Nevertheless, we did our best to find and correct some errors, and we also found professionals to correct them, and underlined the corrected parts for your further review.

   Finally, Fig. 5 caption should state what E2 means. The point you raised is very important and we have amended it in the drawing note.

   The above are the specific questions you raised in our article and our replies.  

    Your argument in the last paragraph raises some constructive questions, which are very enlightening and instructive for us. Transcriptomic analysis of virus-infected cells will get a huge set of databases, and we will get many differential genes. How to conduct further research from these data is worth thinking about. From my point of view, transcriptomics is just a research tool, and we conduct transcriptomic analysis with the aim of finding genes and signaling pathways that reveal the problem we are studying. For example, the purpose of this study was to find out why PRRSV caused pregnant sows to produce stillborn mummified fetuses. Therefore, we selected trophoblast cells that made up the placenta to infect the virus. Our focus was on the genes and pathways that could reveal the mechanisms of reproductive disorders and fetal development obstruction. And the function verification is carried out. We also present genes and pathways beyond our focus to provide ideas and references for interested researchers, greatly expanding researchers' understanding of the pathogenesis of reproductive disorders in PRRSV.The transcriptome of PAM cells infected by PRRSV focuses on the activation of immune responses, such as NFkb signaling pathway, etc. Through our comparison, it was found that the inflammatory factors in trophoblasts did not change significantly. Therefore, different cells have greatly different responses to virus infection due to their different functions.In our paper, in addition to the estrogen signaling pathway mentioned, there are many contents worthy of further study, and we are also making continuous efforts to improve them. In this paper, we only briefly mentioned some directions because there is insufficient experimental data, aiming at stimulating readers' interest in further research and revealing the pathogenesis of PRRSV comprehensively.

    We look forward to hearing from you regarding our submission. We would be glad to respond to any further questions and comments that you may have.

    Kind regards

                                                                                                      Changxu Song

Round 2

Reviewer 2 Report

I thank the authors for answering all my doubts. However, I still believe the research design regarding the Bioinformatics analysis should be revised. A logFC > 0 with a P-Value < 0.05 (not adjusted) leads to many false positive results that compromise the whole understanding of the disease. In addition, the pathways and ontologies enriched are accordingly to what is expected in a viral infection. A more robust bioinformatic analysis could help to prioritize pathways that are relevant, but not commonly studied in the disease.

1. What about the downregulated genes?

2. I did not encountered the full differential gene expression (DGE) tables.

3. Please include the information regarding logFC and P-Value in the Methods section.

Author Response

Dear Editor

  Thank you very much for your letter again. Your suggestions are very helpful to the improvement of our article.In response to your suggestions, we have made modifications in the paper. As for the questions you raised, we would like to make the following explanations:

   P<0.05 is a commonly used differential parameter in high-throughput sequencing, which has been used for differential screening in many reports. Of course, as you said, there was a certain possibility of false positives, but we randomly selected 6 differential genes for fluorescence quantitative pcr test verification, as shown in Figure 3. The variation trend was basically consistent with the results of transcription, which proved that the accuracy of our results was within the acceptable range. Your suggestion to increase the P-value will undoubtedly further promote the accuracy of the results, but it may also lose a lot of important data. Our study was for the purpose of preliminary screening, and obtained the result that PRRSV inhibited the expression of GPER1, a key protein, and carried out functional verification. But this is not the end of the story. We will sift through the key information and have uploaded the raw data to the NCBI database for further research by interested readers. The serial number for SRR16912825, SRR16912824 SRR16912823, SRR16912822, SRR16912827, SRR16912826.

    As you say, the pathways and ontologies enriched are accordingly to what is expected in a viral infection. A more robust bioinformatic analysis could help to prioritize pathways that are relevant. I think your opinion is very instructive, and we will conduct further analysis in the subsequent research. In this paper, P<0.05 is used for analysis, and we are very lucky to find the key factor of PRRSV causing miscarriage of sows. Of course, there will be other factors that need to be further revealed. Your suggestion is very helpful to us.

    1. What about the downregulated genes?

    Answer: PRRSV infected cells can activate many reactions, corresponding to the abnormal expression of many genes, including up-regulated and down-regulated. For example, apoptotic signaling pathway, genes promoting apoptosis will be significantly up-regulated, while those inhibiting apoptosis will be significantly down-regulated, so as to realize the biological process of apoptosis. Our results showed that mRNA was up-regulated in 406 cases and down-regulated in 266 cases, while LncRNA was up-regulated in 284 cases and down-regulated in 192 cases.

     2. I did not encountered the full differential gene expression (DGE) tables.

    Answer: The complete differential gene list is not included in the paper due to the large number. We will upload it in the form of supplementary materials.

     3. Please include the information regarding logFC and P-Value in the Methods section.

     Answer: Thank you very much for your advice, and we have supplemented this part in the article.

     Thank you again for your valuable suggestions on the improvement of our article, and we hope you will be satisfied with our changes and answers. If you still have relevant suggestions and questions, please be sure to contact us, we will do our best to correct.

Kind regards,
Changxu Song
